# Incorporating Rainfall Forecast Data in X-SLIP Platform to Predict the Triggering of Rainfall-Induced Shallow Landslides in Real Time

Michele Placido Antonio Gatto 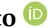

Department of Civil, Environmental, Architectural Engineering, and Mathematics, University of Brescia, Via Branze 38, 25123 Brescia, Italy; michele.gatto@unibs.it

**Abstract:** Extreme and prolonged rainfall resulting from global warming determines a growing need for reliable Landslide Early Warning Systems (LEWS) to manage the risk of rainfall-induced shallow landslides (also called soil slips). Regional LEWS are typically based on data-driven methods because of their greater computational effectiveness, which is greater than the ones of physically based models (PBMs); however, the latter reproduces the physical mechanism of the modelled phenomena, and their modelling is more accurate. The purpose of this research is to investigate the prediction quality of the simplified PBM SLIP (implemented in the X-SLIP platform) when applied on a regional scale by analysing the stability of rain forecasts. X-SLIP was updated to handle the GRIB files (format for weather forecast). Four real-time predictions were simulated on some towns of the Emilia Apennines (northern Italy) involved in widespread soil slips on 5 April 2013; specifically, maps of factors of safety related to this event were derived assuming that X-SLIP had run 72 h, 48 h, 24 h and 12 h in advance. The results indicated that the predictions with forecasts (depending on the forecast quality) are as accurate as the ones derived with rainfall recordings only (benchmark). Moreover, the proposed method provides a reduced number of false alarms when no landslide was reported to occur in the whole area. X-SLIP with rain forecasts can, therefore, represent an important tool to predict the occurrence of future soil slips at a regional scale.

**Keywords:** rainfall-induced shallow landslides; SLIP model; real-time instability predictions; landslide risk management



## 1. Introduction

Global warming and climate change are directly affecting the pattern of precipitation on our planet [1,2]. In recent times, intense rainfall occurred increasingly and triggered other natural phenomena, among which the rainfall-induced shallow landslides (also called soil slips) [3–6]. It deals with slope instabilities due to changes in soil saturation following rainfall infiltration. When unsaturated soils reach full saturation, they experience a loss in strength that can cause a sudden triggering of soil slips under specific geomorphological conditions [7–11]. Soil slips of even limited sizes could cause direct and indirect damage, such as life losses, disruption of the built environment, road closures, etc. [12,13]. Hence, the need for the identification of susceptible areas where to introduce mitigation measures, e.g., based on vegetation [14–16] or for the real-time prediction of phenomena occurrence to raise alarm and respond to emergency situations in time. The latter aspect is the focus of the so-called Landslide Early Warning Systems (LEWS), which detect hazardous events in advance thanks to space–time predictive models and are adopted for civil protection purposes. Relevant reviews on this topic can be found in [17–24]. LEWS are classified according to the scale of application: slope, catchment or regional; in the following, regional LEWS are referenced.

Most regional LEWS provide space–time information via data-driven methods, e.g., based on rainfall thresholds (empirical relationships between rainfall and landslide occur-

rence derived from past events) [25–28] or on artificial intelligence [29–33]. The mentioned methods are applied for real-time predictions thanks to the rain forecast. Recent progress was made by including non-landslide data when developing the data-driven models to reduce the false alarms, i.e., positive points according to the adopted method but no real occurrence [34–37].

Other than data-driven methods, PBMs for the prediction of the triggering of rainfall-induced shallow landslides are formulated on the understanding of the physical problem and the soil's mechanical and hydraulic behaviour [38–42]. Slope stability is commonly analysed using the Limit Equilibrium Method (LEM), combining the statics with hydrological models for the effects of infiltrated rainfall on soil shear strength. For shallow landslides, one-dimensional (1D) LEMs are widespread, based on the infinite slope scheme; literature presented several 1D-LEMs implementing hydrological models of different complexity [43–50]. However, the computational demand makes the application of sophisticated PBMs prohibitive over large areas and, therefore, for regional LEWS. To this purpose, a compromise of the modelling complexity needs to be achieved.

One of the most effective PBM, even simplified, is SLIP (Shallow Landslides Instability Prediction); its simplification is on the time-dependency of soil saturation on rainfall: saturation immediately increases because of infiltration (static rate) but then decreases due to runoff (dynamic rate). Despite its simplicity, SLIP showed interesting predictive skills in different territories and at different scales of analysis [51–57]. It was even adopted by DEWETRA, i.e., the multi-risk platform of the Italian Department of Civil Protection, for pre-alert purposes (analyses with recorded rainfall related to past events) [54]. Gatto and Montrasio [58] presented a MATLAB platform named X-SLIP, where the SLIP model is implemented, and stability can be assessed over large areas with easily available territorial data. However, until now, X-SLIP showed its effectiveness when applied with recorded rainfall; to use it in regional LEWS, it must be enabled to perform real-time predictions of future events, with stability analyses based on rainfall forecasts.

How should X-SLIP include rainfall forecasts to perform real-time stability analyses? What is the relationship between the forecast time and the prediction accuracy? This study answers these questions via the development of a feature in the X-SLIP platform specific for such space predictions in real time. Section 2.1 briefly summarises the SLIP model and X-SLIP platform, while Section 2.2 describes how GRIB data (a common format for weather forecast) are imported and managed by X-SLIP. Section 3 shows the validation of the new feature on an area of Emilia Apennines (northern Italy), where some soil slips were triggered after intense and prolonged rainfall on 5 April 2013; in this area, X-SLIP already demonstrated its prediction capabilities when applied with recordings only [58,59]. Real-time hindcasts are, therefore, conducted on 5 April 2013 at 12 p.m. (survey time of the reference events), assuming to have run X-SLIP 12, 24, 48 and 72 h in advance. To this aim, rain forecasts provided by the Italian Air Force Weather Service are used. Section 4 presents the factors of safety and the prediction qualities computed with forecasts in comparison with the ones related to only recordings (Section 4.1). Finally, Section 4.2 presents a real-time prediction conducted on 2 April 2013 at 12 a.m., a time event when rainfall was comparable to the one which triggered the soil slips on 5 April, but there is no evidence of occurred phenomena; this allows us to assess the effectiveness of X-SLIP real-time predictions even on non-landslide days.

## 2. Materials and Methods

### 2.1. The SLIP Model and X-SLIP Platform

SLIP (Shallow Landslide Instability Prediction) is a simplified physically based model formulated by Montrasio [44] to predict the triggering of soil slips. Such phenomena are typically characterised by planar failure surfaces established at the depth of the topsoil, i.e., the shallow soil characterised by extensive macropores where the rainfall infiltrates [60,61];

coherently, SLIP evaluates the one-direction factor of safety $F_S$ of the topsoil via the infinite slope scheme, as follows:

$$F_S(t) = \frac{\tan \varphi\prime}{\tan \beta} + \frac{2 \cdot [c' + c_\psi(t) + c_r]}{\gamma H \sin 2\beta} \tag{1}$$

According to Equation (1), $F_S$ depends on morphology (slope angle $\beta$), the thickness $H$, and the physical and mechanical parameters of the topsoil (unit weight $\gamma$, effective cohesion $c'$, apparent cohesion $c_\psi$, and friction angle $\varphi'$). The term $c_r$ has recently been introduced by Montrasio et al. [59] to model the root reinforcement provided by vegetation.

As it is known, the topsoil's stability is physically related to the soil's apparent cohesion $c_\psi$, which varies with time depending on rainfall and is zero when full saturation is reached. SLIP evaluates $c_\psi$ via a time-dependent relationship experimentally derived by Montrasio [44] as follows:

$$c_\psi(t) = AS_r (1 - S_r)^\lambda [1 - m(t)]^\alpha \tag{2}$$

$S_r$ is the soil's degree of saturation, while $A$, $\lambda$ and $\alpha$ are modelling parameters. $m(t)$ is the rainfall-dependent variable, in which its expression is based on some simplification; it represents the thickness of the fully saturated horizontal layer in the topsoil, having an area equivalent to the saturation bubbles created after rainfall infiltration. Considering the rate $1 - \beta^*$ of rainfall $h_j$ infiltrating the topsoil, $m(t)$ is computed as follows:

$$m(t) = \frac{1 - \beta^*}{nH(1 - S_r)} \sum_{j=1}^{\omega} h_j e^{-k_t(t-t_j)} \tag{3}$$

$n$ is the soil's porosity. $k_t$ is the slope drainage allowing us to introduce the dynamic variation of rainfall $h_j$ (occurred at time $t_j$) in consecutive time steps ($t > t_j$). $\beta^*$ takes into account the reduction of infiltrated rainfall because of steep slopes or plant interception [58,59]. In summary, Equations (2) and (3) consider both the increase in soil's saturation provided by rainfall and the decrease due to runoff. $S_r$ represents an initial saturation and is updated according to the rainfall in a specific window (set of $\omega$ data). Losi [62] demonstrated that a 30-day rainfall window is significant enough to capture some of the occurred events. In such a way, $S_r$ is related to a month before the event when stability is analysed, and the rainfall is collected over a period of 30 days ($\omega$ = 30 days). A detailed description of the SLIP model is presented by [44,51,55,59].

The time-dependent SLIP formulation was recently implemented in the MATLAB fully implemented platform X-SLIP [58]. The platform is based on a multi-approach algorithm consisting of different techniques allowing us to evaluate the space–time stability with all the parameters required by SLIP derived from easily available territorial data (Figure 1). First, the reference grid, i.e., the group of pixels where SLIP is applied, comes from the Digital Terrain Model (DTM). These territorial data are also used to build the matrix of slope angles using the MATLAB *gradientm* function, which is based on finite difference method and Lagrange interpolation. Matrices of soil parameters (user defined) can include homogeneous parameters (approach suggested for small areas) or lithology-dependent parameters, varying with space according to the lithology map (or the map of soil texture, when available). In the latter case, a problem of point-in-polygon is solved using the *inpoly* function, which identifies the lithology polygon, including each pixel [58,63,64]; a set of soil parameters is then assigned with each lithology type according to a user-defined association. A similar approach is followed for the matrices of vegetation parameters, built according to the vegetation map and user-defined set of species-dependent parameters. Finally, rainfall matrices must be created to apply the simplified hydrological model. Until now, only data recorded by rain stations have been considered. They are related to unregular spaced points and distributed via interpolation on the pixels of the reference grid. To this purpose,

the Natural Neighbour algorithm is adopted [65] and implemented in MATLAB using the *scatteredInterpolant* function.

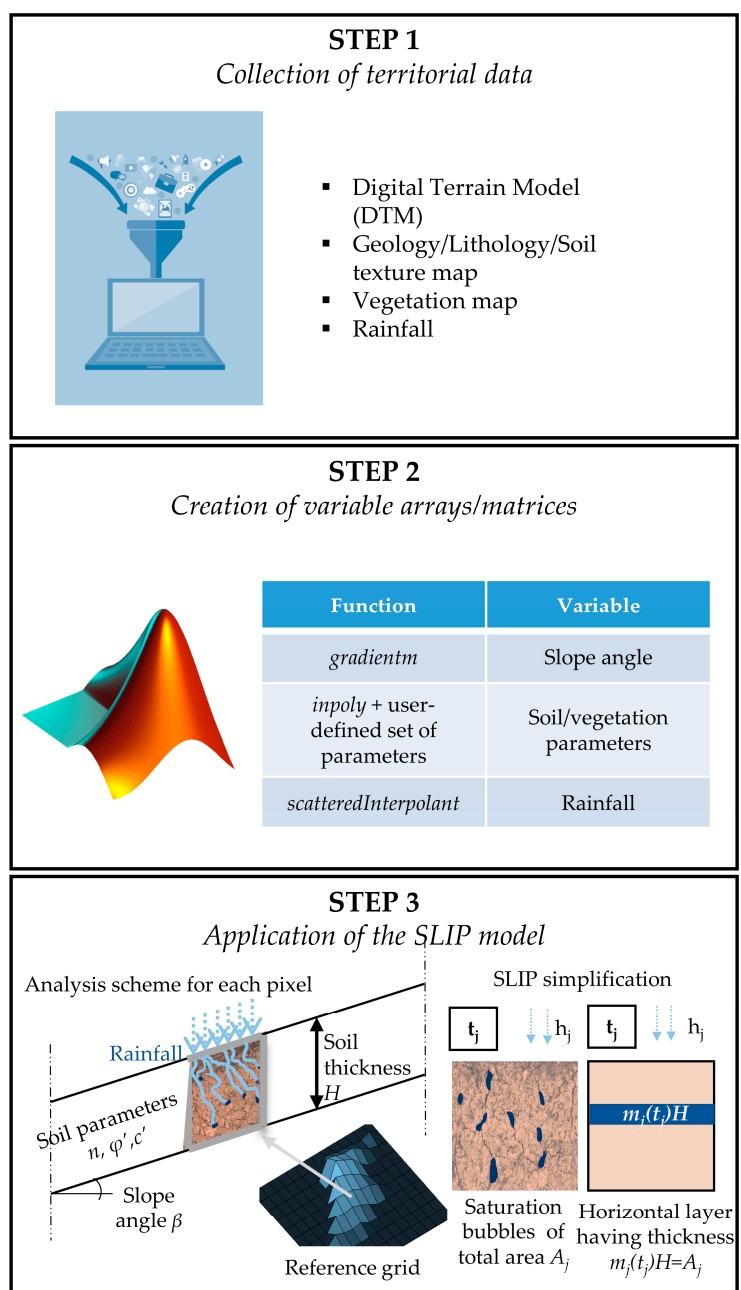

**Figure 1.** Steps followed by the X-SLIP platform to apply the SLIP model over large areas.

*2.2. A New Feature of X-SLIP Platform for Real-Time Prediction through Rainfall Forecasts*

X-SLIP platform could be implemented in any LEWS if stability of the topsoil were analysed in real time; this is possible by integrating rain forecast in SLIP runs so that any critical situation can be predicted well in advance. In Europe, weather forecasts are based on the ECMWF (European Centre for Medium-Range Weather Forecasts) model, considered as one of the most reliable [66]. Some countries, among which Italy, adopt COSMO (COnsortium for Small-scale MOdeling); based on ECMWF model, COSMO integrates details with higher resolution and allows us to predict localised weather events [67]. Both ECMWF and COSMO data are stored in GRIB2 (.grib) file. To this study's aim, how such data are contemplated in X-SLIP is here described. The description is related to COSMO, being the study area where this research is validated in Italy.

MATLAB (development environment for X-SLIP) allows us to import the .grib files using the *ncdataset* function, included in the NCToolbox; such function returns a Netcdf (Network Common Data Form) Dataset, with variables included in it. The ones of major interest are the forecast times, the forecast values and the coordinates of points to which predictions are related. Typically, COSMO performs two runs every day, one at 00 UTC and another at 12 UTC. For each jth run, forecasts are hourly provided up to 15 days later; in this study, a maximum forecast window of 72 h (three days) is considered, being this time gap is commonly recognised as accurate for weather forecasts [68]. When imported in X-SLIP, the jth GRIB file, therefore, returns a 72-by-m-by-n matrix, with data of 72 forecast events (hourly spaced) evaluated at the points of an m-by-n grid. Grids considered by COSMO have a resolution lower than the morphological data of the national/regional database, representing the reference grid for X-SLIP. Interpolation is, therefore, performed to estimate values of rainfall predictions at the X-SLIP grid; for this purpose, the *scatteredInterpolant* function is applied (see Section 2.1). Note that the COSMO mesh must completely cover the study area to obtain reliable interpolations through *scatteredInterpolant* [58]. The procedure of import and interpolation of GRIB data is schematised in Figure 2.

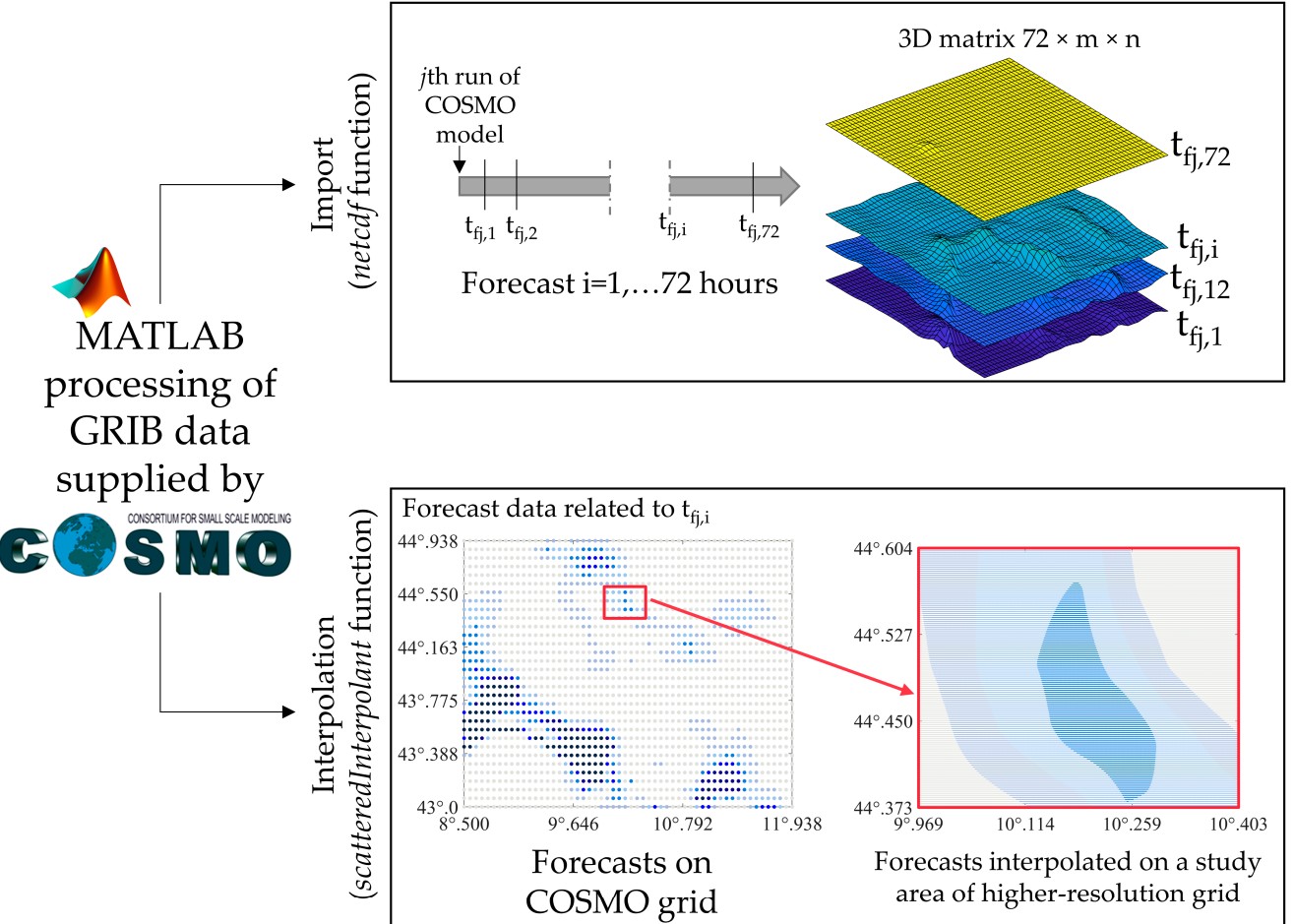

**Figure 2.** Integration of GRIB files in X-SLIP platform: import (*netcdf* function) and interpolation (*scatteredInterpolant* function).

As previously discussed, SLIP analyses are based on a simplified hydrological model, expressed by Equation (3); this evaluates the soil's saturation varying with time depending on the rainfall occurred on a rainfall window *RW* preceding the generic event $e_P$ when stability is analysed (*RW* = 30 days, according to [62]). For real-time predictions of future events, such *RW* must include forecasted data, as well as recorded ones; the final structure of the rainfall array is illustrated in Figure 3. Note that it deals with a "cell" array, storing

matrices in each position. To investigate the influence of the forecast window in X-SLIP real-time predictions of past events, the final part of the rainfall array must be, therefore, filled with forecast data $i$ of the $j$th COSMO run suitably selected (rain forecast from $t_{fj,1}$ to $t_{fj,i}$, with $i$ depending on the chosen forecast window).

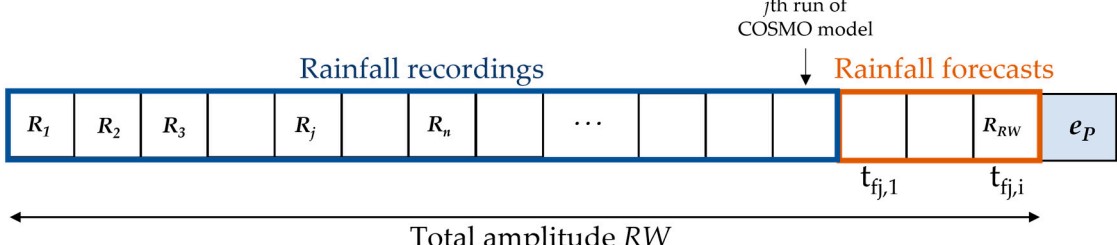

**Figure 3.** Scheme of the rainfall array used in X-SLIP for real-time stability analysis at the event $e_P$.

The approach here described can be adapted for hourly predictions by moving properly the *RW* array and updating the m value accordingly (See Equation (3)).

### 3. Application of the Updated X-SLIP Platform in Some Towns of the Emilia Apennines (Northern Italy) Involved by Soil Slips on 5 April

*3.1. Case Study Description*

The Emilia Apennines, i.e., the northern Apennines located in the Emilia Romagna region (north of Italy), are particularly landslide-prone due to the weak lithology (mainly arenaceous and calcareous flysch, with widespread clayey components) and the presence of groundwater [69]. In April 2013, about 1500 events were detected in this area, triggered after a long and intense rainy period from March to the first ten days of April; such phenomena caused major inconveniences, such as damage to crops, settlements and road disruptions, so that a state of emergency was declared to guarantee the essential level of public services. Based on well-recognised classifications, types of rainfall-induced shallow landslides can be debris slides or debris/earth flow. In the former, slides occur along only one interface, whilst in the latter, there are relative movements in numerous layers of the involved mass. The SLIP model, based on the infinite slope scheme, is suitable to analyse, with good approximation, the initiation mechanism of all the mentioned types. To validate the proposed update of the X-SLIP platform, real-time predictions are simulated on four of the involved towns, all located in the Parma province: Neviano degli Arduini, Corniglio, Tizzano Val Parma and Palanzano. There, Terrone [70] detected 43 soil slips on 5 April 2013 at 12 p.m.; such information is used to evaluate the prediction quality of X-SLIP hindcasts. Figure 4 shows the study area and the positions of the surveyed phenomena, together with some photos representative of the types of phenomena that occurred.

*3.2. Real-Time Hindcasts on April 2013 through X-SLIP*

X-SLIP is applied to space predictions in real time, performed some days preceding the events that occurred on 5 April 2013. For the spatial application of the SLIP model, the morphological, lithological and vegetation data are downloaded from the open-source database of the Emilia Romagna region. Territorial limits and the DTM have projected system EPSG 25832 (ETRS 89 UTM 32N), while the Lithology and Vegetation Map is in EPSG 32632 (WGS84 UTM 32N). EPSG 25832 is considered a reference system for all the analyses. 23 tiles (each of extension 7320 m × 6280 m and resolution 5 m × 5 m) of the regional DTM cover the area in question; to enhance the computational effectiveness, the resolution is reduced to 20 m, and only pixels within the territorial boundaries of the towns are taken. Overall, stability is analysed on more than one million pixels (1,319,976). From elevations included in the DTM, the array of slope angles is derived (Figure 5a). Soil parameters are assigned based on the lithology map illustrated in Figure 5b. A set of parameters coming from previous back analyses is associated with each lithological

class [47]; Table 1 summarises the user-defined lithology–soil parameters association. The topsoil's thickness $H$ is set constant to 1.2 m for the whole study area, being the average observed during the occurred phenomena [70]. The initial degree of saturation $S_r$ is set to 0.8 for the whole study area; this is a seasonal value typical of previous experimental observations in this area [47]. Regarding the vegetation parameters, the starting point is the vegetation map shown in Figure 5c; to each vegetation species, literature values of root cohesion $c_r$ and rainfall interception $\beta^*$ are attributed [59]. These are reported in Table 2.

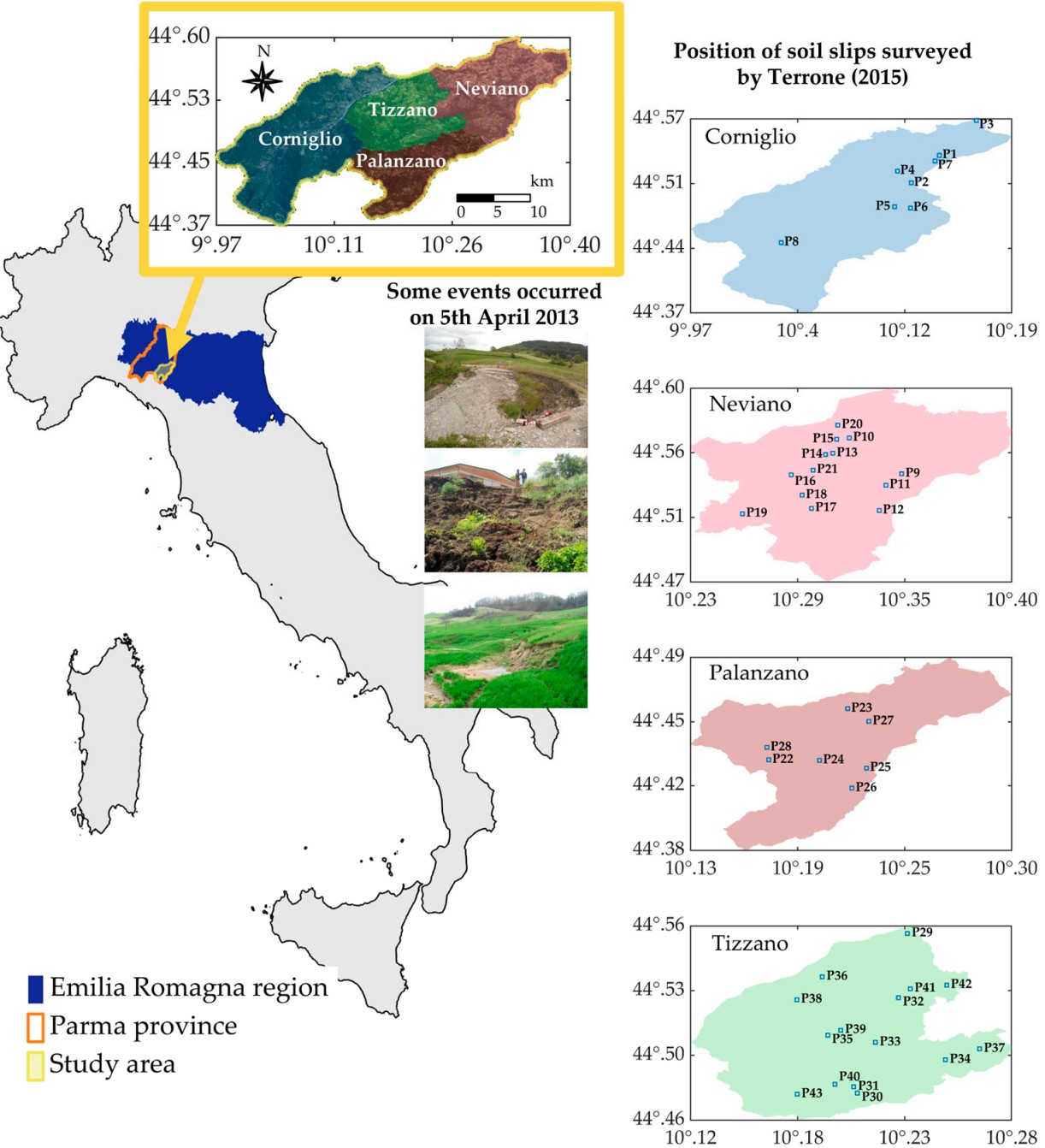

**Figure 4.** Geographical framework of the study area, some photos of the events that occurred on 5 April 2013, and the positions of soil slips detected by Terrone [70].

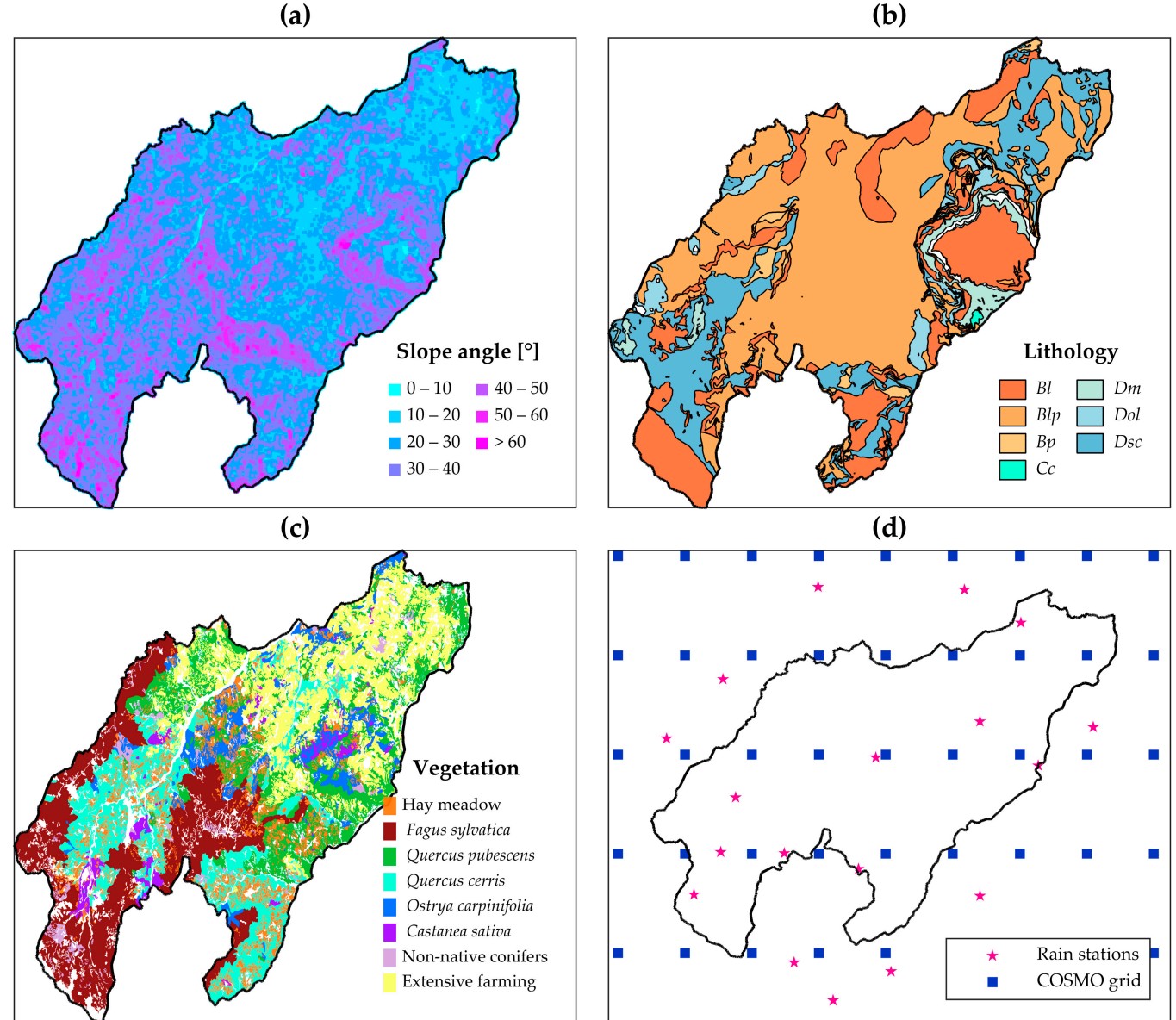

**Figure 5.** Raw data for X-SLIP analyses. (**a**) Slope angles; (**b**) lithologies; (**c**) vegetation species; and (**d**) points of COSMO grid and rain stations, respectively, related to forecasted and recorded rainfall.

**Table 1.** Lithology–soil type association, after [47].

| Lithology Abbreviation | Description | Soil Type | $n$ (-) | $\varphi'$ (°) | $c'$ (kPa) | $k_t$ (h$^{-1}$) | $A$ (kPa) |
|---|---|---|---|---|---|---|---|
| Bl | Rocks made up of alternation with stone dominant levels | Sand | 0.40 | 35 | 0 | 0. 0180 | 40 |
| Blp | Rocks made up of an alternation of stone and pelitic levels | Sand | 0.40 | 35 | 0 | 0. 0180 | 40 |
| Bp | Rocks made up of alternation with pelitic dominant levels | Sand | 0.40 | 35 | 0 | 0. 0180 | 40 |
| Cc | Conglomerates and clast supported breccias | Sand | 0.40 | 35 | 0 | 0. 0180 | 40 |
| Dm | Marlstone | Deposits | 0.45 | 30 | 5 | 0.0250 | 40 |
| Dol | Claystone breccias | Clayey silt | 0.46 | 25 | 0 | 0.0180 | 80 |
| Dsc | Scaly clays | Clay | 0.50 | 20 | 10 | 0.0007 | 100 |

**Table 2.** Vegetation parameters adopted for each species, according to [59].

| Vegetative Species | $c_r$ (kPa) | $\beta^*$ (-) |
|---|---|---|
| Hay Meadow/Extensive Farming | - | 0.40 |
| *Fagus sylvatica* | 21.26 | 0.30 |
| *Quercus pubescens* | 0.2 | 0.17 |
| *Quercus cerris* | 25.9 | 0.21 |
| *Ostrya carpinifolia* | 14.4 | 0.20 |
| *Castanea sativa* | 14.0 | 0.45 |
| Non-native conifers | 0.2 | 0.30 |

Four hindcasts of real-time predictions are conducted; the term "hindcast" is used because the real-time analysis is referred to a past event. Being the occurrence of the 43 phenomena associated with 5 April 2013 at 12 p.m., stability at that point in time is investigated assuming to have run the analyses: (i) 12 h; (ii) 24 h; (iii) 48 h; and (iv) 72 h in advance of the occurred event. Each case corresponds to a specific run of the COSMO model and a different forecast window; the forecast data have been provided by the Italian Air Force Weather Service. As discussed, SLIP requires rainfall to fall 30 days before the stability analysis is performed. Other than rain data in the forecast window immediately preceding the analysis event, rain recordings are considered for the remaining part of the rainfall window (see Figure 3). Hourly rainfall data, recorded at 18 stations surrounding and inside the study area, has been supplied by Dexter Arpae; recordings from 4 March 2013 to the date of the COSMO model running (depending on the forecast case) are adopted. Figure 5d shows the positions of the rain stations and the points of the COSMO mesh to which the provided rain forecasts are related. Table 3 summarises the cases investigated, the times of COSMO run, the forecast and the recording hours (rainfall window of total amplitude 720 h, i.e., 30 days by 24 h).

**Table 3.** Summary of the simulated real-time predictions.

| Case Analysed | Run of COSMO Model | Forecast Hours | Recording Hours |
|---|---|---|---|
| 12 h | 5 April 12 a.m. | 12 | 708 |
| 24 h | 4 April 12 p.m. | 24 | 696 |
| 48 h | 3 April 12 p.m. | 48 | 672 |
| 72 h | 2 April 12 p.m. | 72 | 648 |

## 4. Results

### 4.1. Cumulative Rainfall of Forecasts and Recordings

Figure 6 shows a comparison between the forecast and recorded precipitation, interpolated on the reference grid and cumulated in the forecast windows of amplitude 72 h (Figure 6a,b), 48 h (Figure 6c,d), 24 h (Figure 6e,f) and 12 h (Figure 6g,h). In almost all cases, recordings are greater than the forecasts over the whole study area. An exception is made for the recorded rainfall cumulated 12 h before the reference event, which is smaller in some zones. To better visualise the differences, the relative error *RE* between forecast $h_{w,f}$ and recorded rain $h_{w,r}$ is computed, assuming the latter as reference:

$$RE = \frac{h_{w,f} - h_{w,r}}{h_{w,r}} \tag{4}$$

Maps of *RE* for the four cases are illustrated in Figure 7. In 24 h and 12 h cases, there are areas where *RE* is greater than 1 (yellow to red) (Figure 7c,d); there, reduced values of factor of safety are expected from X-SLIP predictions.

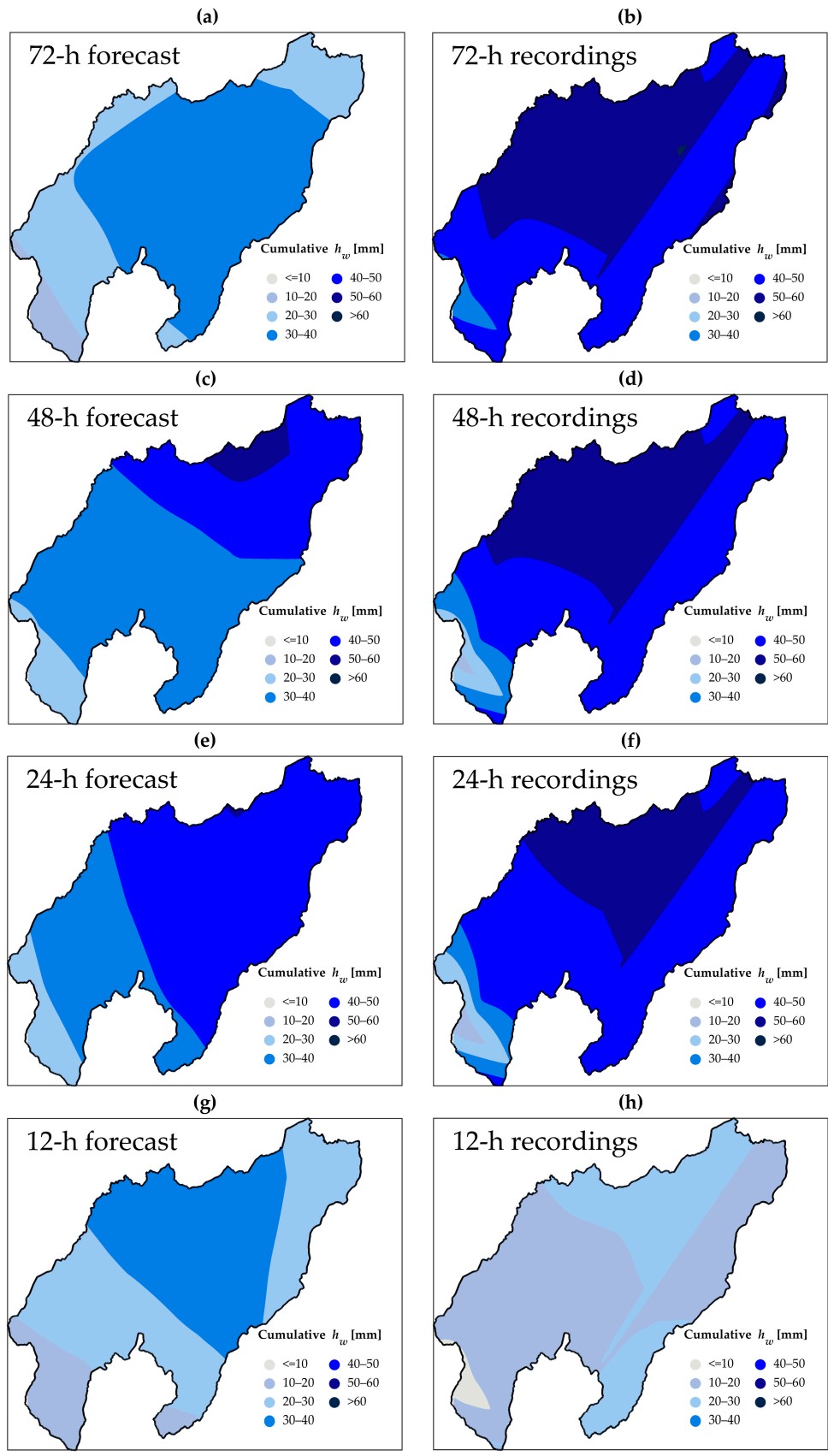

**Figure 6.** Comparison of forecast and recorded rainfall cumulated on: (**a**) and (**b**) 72 h; (**c**) and (**d**) 48 h; (**e**) and (**f**) 24 h; (**g**) and (**h**) 12 h.

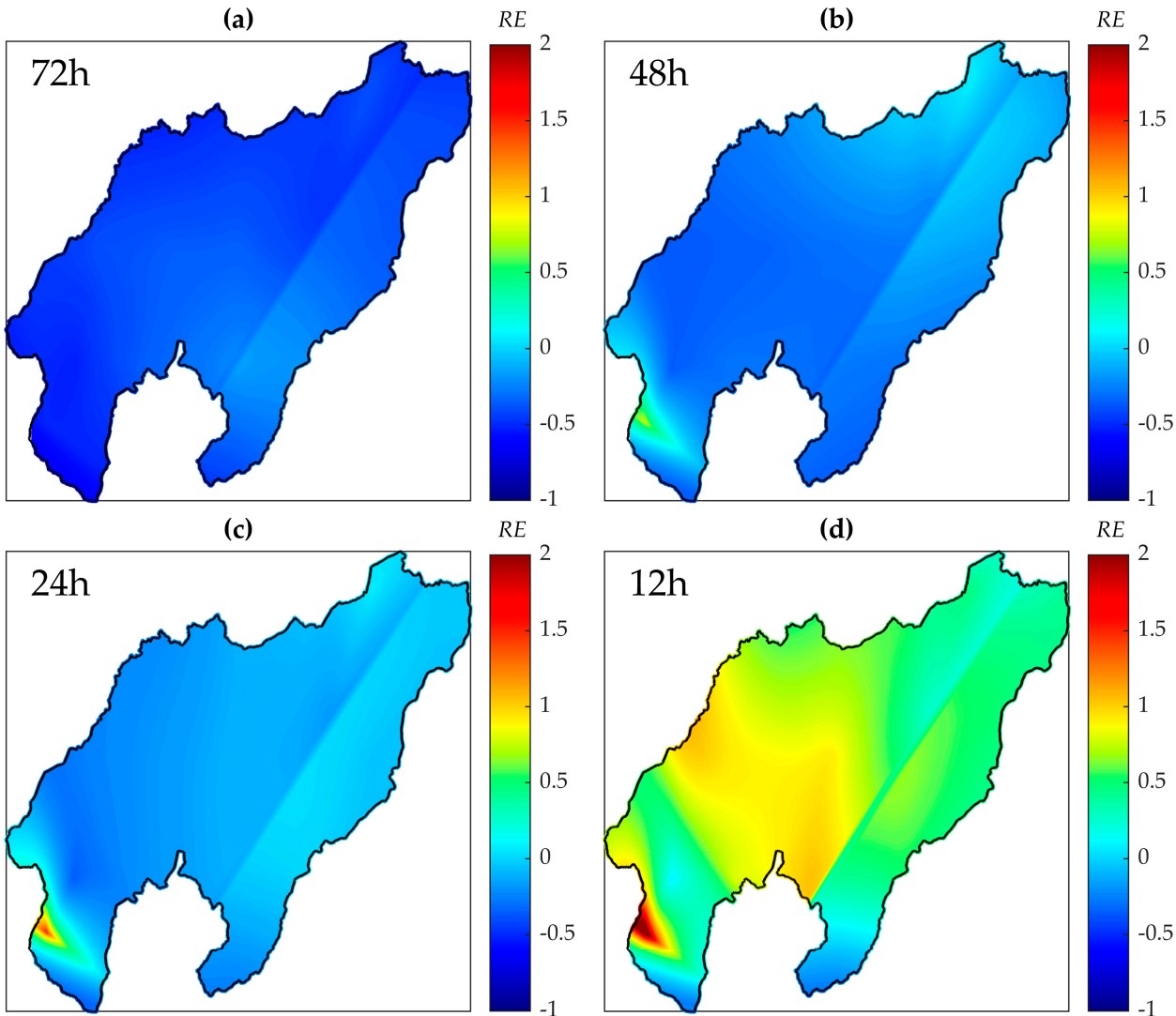

**Figure 7.** Relative Error (*RE*) evaluated between the forecast and recorded rainfall (reference value) cumulated on (**a**) 72 h; (**b**) 48 h; (**c**) 24 h and (**d**) 12 h before the occurred event.

*4.2. Factor of Safety on 5 April 2013 at 12 p.m.*

Figure 8 reports the results of stability analyses performed via X-SLIP on 5 April 2013 at 12 p.m., grouped considering $F_S$ less than 1 (unstable pixels), between 1 and 1.5 (critical pixels) or greater than 1.5 (stable pixels); soil slip points surveyed by Terrone [70] are also plotted for a qualitative comparison. The target outcome is represented by factors of safety $F_S$ evaluated with the recordings only (Figure 8a); as already observed by Montrasio et al. [59], in this case, almost all the detected instabilities are captured (pixels with $F_S$ less than 1). Figure 8b–e illustrate the maps obtainable if X-SLIP had been run on: 2 April 12 p.m. (72 h before, Figure 8b), 3 April 12 p.m. (48 h before, Figure 8c), 4 April 12 p.m. (24 h before, Figure 8d), 5 April 12 a.m. (12 h before, Figure 8e). In some areas, instabilities could be efficiently predicted three to two days before the occurred event; however, the most encouraging result is derived a day before, when predictions are surely very close to the target result for the whole study area. This means that for the case in question, 24 h would have been adequate to enable authorities to raise an alert or take the necessary steps to avoid inconvenience.

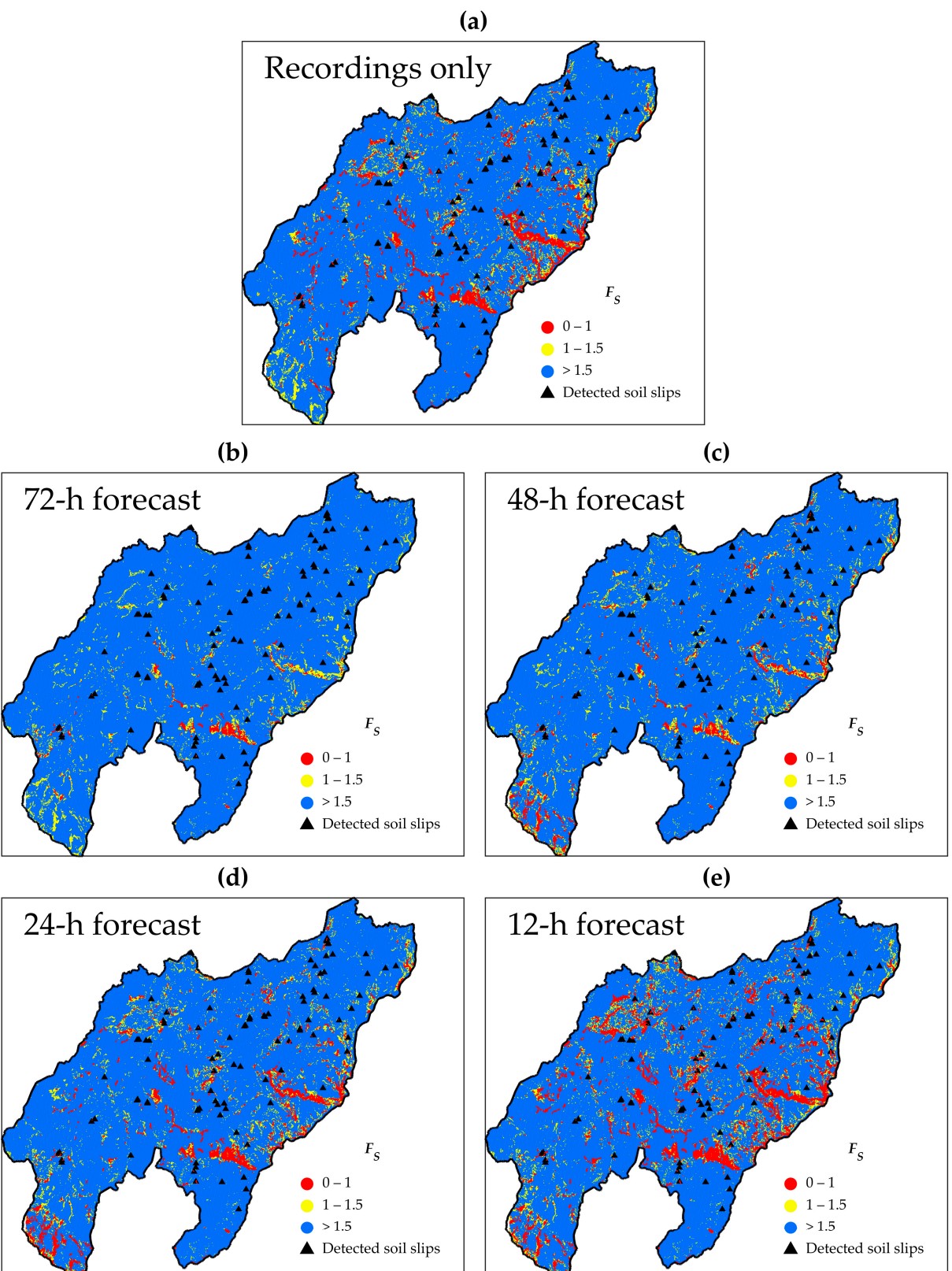

**Figure 8.** Maps of $F_S$ evaluated on 5 April 2013 at 12 p.m. with (**a**) only recorded rainfall (X-SLIP run after the event occurrence); (**b**) recorded and 72 h forecast rain (X-SLIP run three days before the occurrence); (**c**) recorded and 48 h forecast rain (X-SLIP run two days before the occurrence); (**d**) recorded and 24 h forecast rain (X-SLIP run one day before the occurrence); (**e**) recorded and 12 h forecast rain (X-SLIP run 12 h before the occurrence).

In Figure 9, the amount of unstable, critical and stable pixels is shown for each case; specifically, unstable pixels are 28,752 with recordings only, 25,714 with 24 h forecasts and 43,033 with 12 h forecasts. This evidences that X-SLIP with forecasts is as effective as with recordings only, provided that forecasts are accurate. The greater number of unstable and critical pixels with 12 h forecasts reflects the result shown in Figure 7, where forecasted rain is demonstrated to be higher in the whole study area.

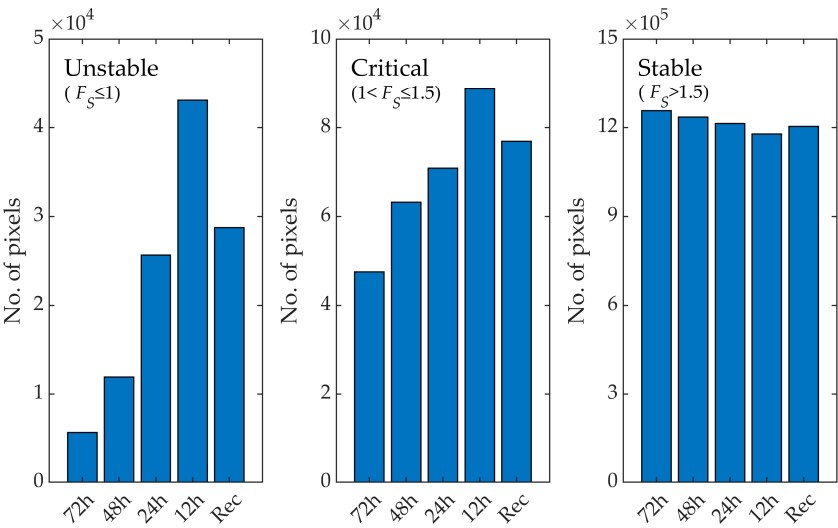

**Figure 9.** Amount of unstable ($F_S \leq 1$), critical ($1 < F_S \leq 1.5$) and stable ($F_S > 1.5$) pixels derived from X-SLIP analyses on 5 April 2013 at 12 p.m.

### 4.3. ROC Analysis from Predictions on 5 April 2013 12 a.m.

The prediction quality of X-SLIP real-time hindcasts is here quantitatively investigated through the ROC (Receiver Operating Characteristic) curve [71]. This is based on the concepts of True Positive Rate *TPR* and True Negative Rate *TNR*, both evaluated by knowing the positions of soil slips detected by Terrone [70]. Specifically, according to X-SLIP results, a pixel is: (i) "positive" if $F_S$ is less than or equal to a threshold value $F_{S,TH}$; (ii) "negative" if $F_S$ is greater than $F_{S,TH}$. Such classification is "true" or "false", depending on the belonging of pixels to areas around the detected points or not. Conventionally, squares 45 m $\times$ 45 m are taken as unstable polygons considering possible errors committed by the inspectors or by the satellite in adverse conditions, as well as the typical extent of the phenomenon under analysis. A positive pixel inside the unstable polygon is identified as True Positive (*TP*), outside as False Positive (*FP*), while a negative pixel is True Negative (*FP*) if outside the unstable polygon and False Negative (*FN*) when inside. *TPR* and *TNR* are therefore computed as follows:

$$TPR = \frac{TP}{TP + FN} \tag{5a}$$

$$TNR = \frac{TN}{TN + FP} \tag{5b}$$

The ROC curve is defined by changing $F_{S,TH}$ from 0 to 70 and deriving the corresponding *TPR*s and *TNR*s; then, the prediction quality is evaluated through the Area Under the Curve (*AUC*), i.e., the integral of the ROC curve. Figure 10 shows the ROC curves for all the cases analysed, together with the corresponding *AUC*. It is interesting to observe that the curves derived from analyses with rain forecasts are quite coincident with the ones with recordings only, and the *AUC* is greater than 80% in all cases. This proves not only X-SLIP accuracy (already clear from previous studies conducted with only rain recordings) but also the stability of the SLIP model itself even in real-time predictions, performed with forecast rain. Note that the *AUC* is maximum with 72 h forecasts because of fewer unstable pixels and consequently *FP*s.

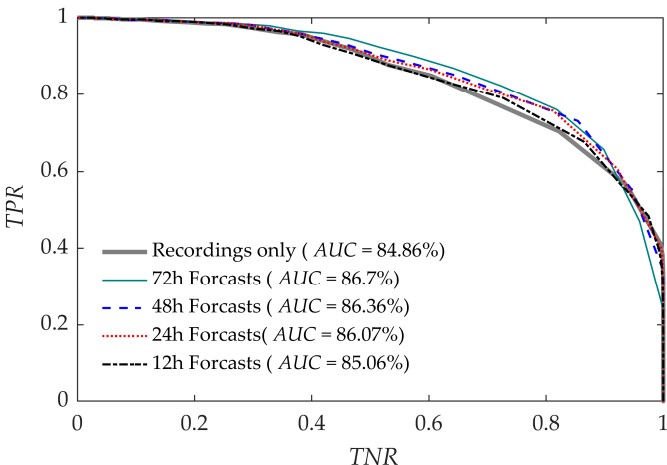

**Figure 10.** ROC curves computed from the results of the four real-time hindcasts based on rain forecasts and the prediction with rain recordings only.

### 4.4. Factor of Safety on a Non-Landslide Event

X-SLIP accuracy in time prediction is, here, investigated on a non-landslide day, i.e., when it is not aware that soil slips were triggered despite intense rainfall. By analysing the rainfall trend shown in Figure 11, related to three rain stations along a diagonal line crossing the study area, a peak comparable with the triggering one (5 April) is noticed on 2 April. Further, X-SLIP hindcasts of real-time predictions are, therefore, conducted on this event, with all data previously discussed, including the cases analysed (Table 1), but with the run of the COSMO model and the dates of rain recordings properly shifted and selected.

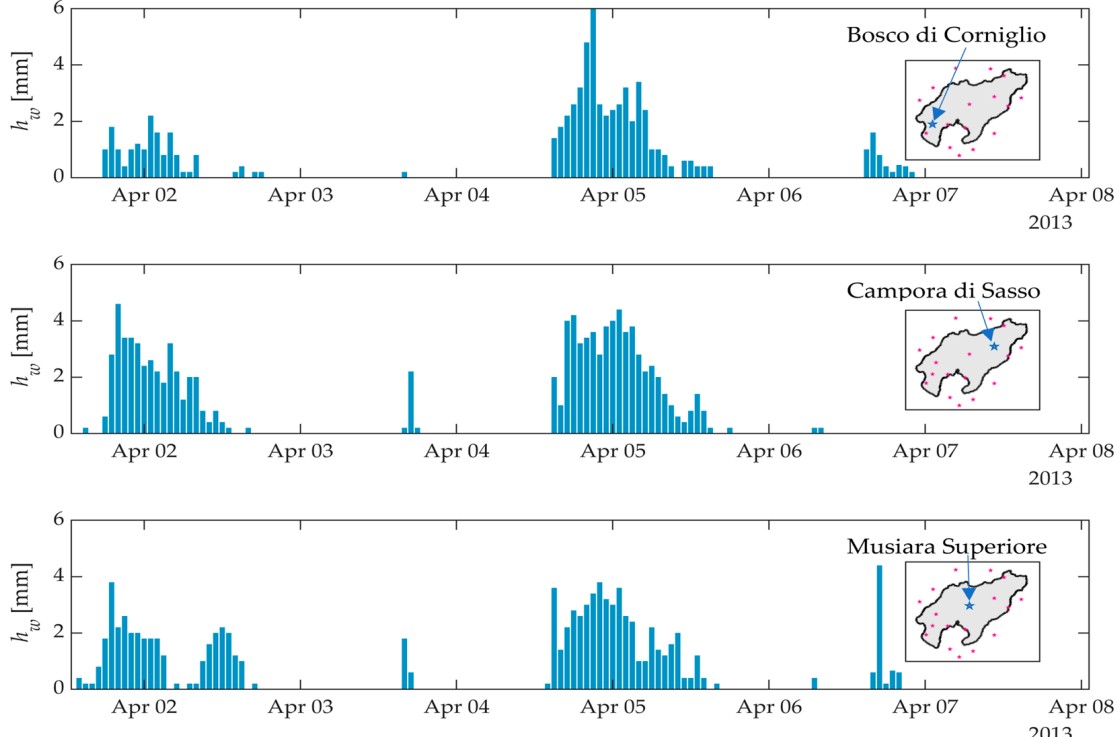

**Figure 11.** Trend of rainfall recorded from 2 to 5 April 2013 at three stations inside the study area.

Maps of the computed factors of safety are shown in Figure 12. It can be observed that in all cases, most of the pixels have $F_S$ greater than 1; this confirms the absence of any report of occurred events. There are small zones occupied by red pixels in the southwest

and the east part, which can be attributed to a bad X-SLIP prediction or could actually have been instabilities not detected because not involving the built environment. The amount of unstable, critical and stable pixels (Figure 13) evidence that the unstable pixels are around 10 k, i.e., less than 1% of the total number of pixels (1,319,976). A majority of pixels are stable, with a quite high percentage of critical pixels due to the temporal proximity of the landslide day. Even in this case, there is a good agreement between the space-time prediction derived via X-SLIP a posteriori (with recordings only) or a priori (with forecasts) for early warning purposes.

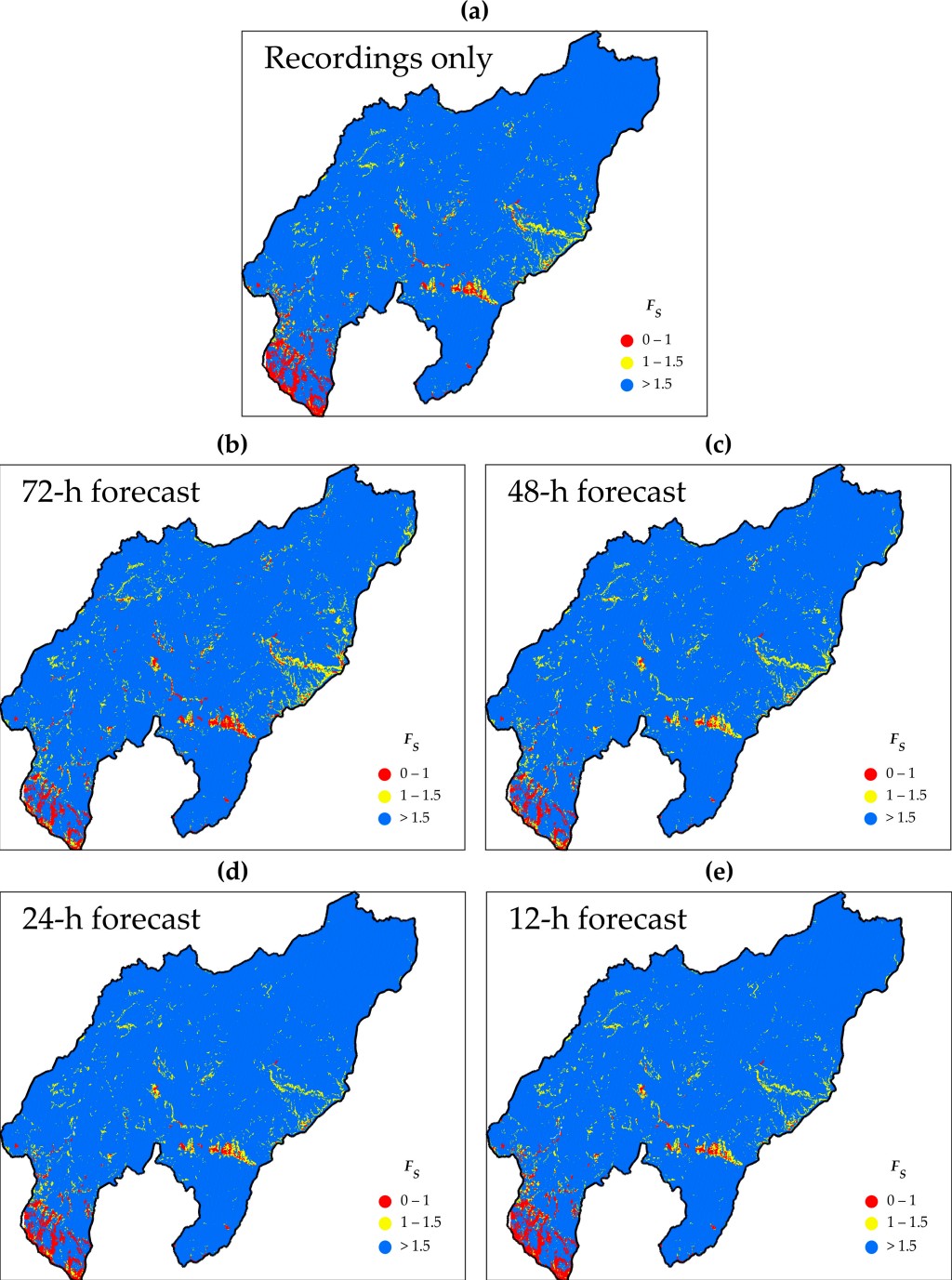

**Figure 12.** Maps of $F_S$ evaluated on 2 April 2013 at 12 a.m. with (**a**) only recorded rainfall; (**b**) recorded and 72 h forecast rain; (**c**) recorded and 48 h forecast rain; (**d**) recorded and 24 h forecast rain; (**e**) recorded and 12 h forecast rain (X-SLIP run 12 h before the occurrence).

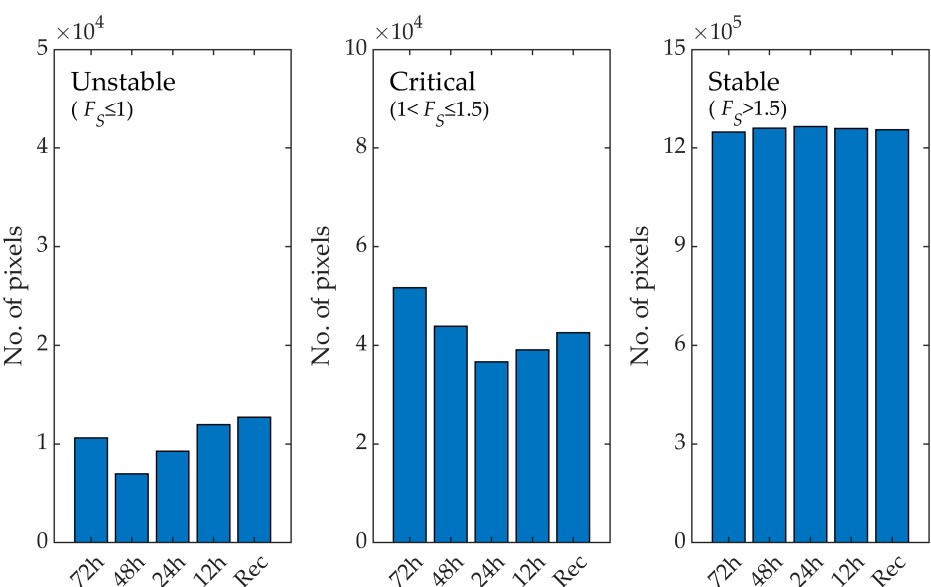

**Figure 13.** Amount of unstable ($F_S \leq 1$), critical ($1 < F_S \leq 1.5$) and stable ($F_S > 1.5$) pixels derived from X-SLIP analyses on 2 April 2013 at 00 a.m.

## 5. Discussion

The effects of global warming and climate change are more serious and more intense than they have ever been. One of the major consequences of extreme rainfall, amongst others, is the increasingly frequent triggering of shallow landslides. It follows that the prediction and mitigation of such phenomena are highly important. Although the analyses of physically based models (PBMs) are more reliable, PBMs are mainly adopted in susceptibility assessment [38,72], being statistical approaches based on rainfall thresholds predominant in regional Landslide Early Warning Systems (LEWS) [25–28]. The results of this study have shown that SLIP, i.e., simplified PBMs implemented in the X-SLIP platform, can provide accurate real-time predictions when the stability analyses integrate rainfall forecasts.

The demonstrated prediction quality is totally in line with previous studies based on such a model [47,58,59]; the absolute novelty is that such quality is now derived from forecasted rainfall, making this model and the related platform suitable for real-time predictions in regional LEWS. Compared to Ho et al. [50], who proposed similar analyses with another PBM, X-SLIP accuracy is even better (*AUC* greater than 80%), and the "lead time", i.e., the time in advance for raising alerts, is longer (24 h in this study, 6 h in [50]). X-SLIP lead time is in good agreement with [18], who proposed a model based on 24 h imminent rainfall. It is worth noting that such authors included the effects of antecedent rainfall in their analysis, as also carried out by [28], but with a statistically derived soil saturation. Although this can be assimilated to SLIP, statistical correlation cannot be equal to a hydrological model, even simplified. SLIP formulation considers directly the soil's hydraulic permeability, and this allows us not only to have a more general model (just select the proper parameter) but also to overcome the uncertainty on the antecedent days of rainfall affecting saturation, being the 30 days included in the SLIP analyses representative enough.

The encouraging result of this study enables X-SLIP analyses based on rainfall forecasts to be adopted in regional LEWS. This is also supported by the computation times of X-SLIP discussed in [58]. The rainfall interpolation is the slowest algorithm, but the reported time to derive 720 rainfall maps (24 hourly maps for 30 days) in the study area of this research is 1925.83 s (with a common laptop equipped with AMD Ryzen 7 4700U, Radeon Graphics 2.00 GHz, 16 Gb RAM and no dedicated GPU). It deals with a very low computation time for a PBM, if compared to others: e.g., TRIGRS requires about 12 min to simulate the

infiltration of a single rainfall in one million pixels [73]. By imaging to use X-SLIP in LEWS for hourly/daily real-time predictions, most of the 30-day antecedent rainfall recordings will be already interpolated; it will be just the import and interpolation of 72 h forecasts (provided twice a day by the COSMO model) that take only a few minutes. This confirms the suitability of the proposed method for civil protection purposes.

## 6. Conclusions

This study highlighted the effectiveness of the SLIP model when applied to predict the space triggering of soil slips in real time. For this purpose, the X-SLIP platform was updated to handle rainfall forecasts in GRIB format and to evaluate the effect of the infiltrated rainfall on soil saturation from both forecasted and recorded rainfall. The main considerations deduced by the results of this study can be summarised as follows.

- X-SLIP analyses based on the combination of rainfall forecast and recordings are as accurate as the ones based only on recordings. This is observed both in landslide and non-landslide events (when no landslide should have occurred).
- The quality of X-SLIP predictions in real-time is strongly dependent on the quality of rainfall forecasts.
- Most of the instabilities that occurred in the Emilia Romagna region in April 2013 would have been well predicted by X-SLIP 24 h in advance.

Further studies will regard the refinement of the selected soil and vegetation parameters to reduce the number of false alarms and the use of spatially distributed rainfall fields (e.g., radar measurements) instead of rain gauge recordings because the interpolation of the latter is a rough approximation. Moreover, how the reliability of forecasts themselves affects X-SLIP accuracy in different times and areas, as well as how to assign different alert levels depending on the forecast time and the number of unstable pixels identified via X-SLIP will be investigated.

**Funding:** This research received no external funding.

**Data Availability Statement:** The data presented in this study are available on request from the corresponding author. The data are not publicly available because the availability depends on rainfall forecasts owned by the Italian Air Force Weather Service, provided after an official request with authorisation for research and publication use.

**Acknowledgments:** The author would like to thank the Italian Air Force Weather Service (Servizio Meteorologico dell'Aeronautica Militare) for their collaboration in providing the data of rainfall forecasts.

**Conflicts of Interest:** The author declares no conflict of interest.

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
