# Peer review of "Incorporating Rainfall Forecast Data in X-SLIP Platform to Predict the Triggering of Rainfall-Induced Shallow Landslides in Real Time"

_geosciences, doi:10.3390/geosciences13070215_

Round 1
Reviewer 1 Report
Very interesting paper and documented analysed cases.
The original proposed procedure (X-SLIP) can represent an important tool to predict possible occurrence of soil slips.
Author Response
Dear Reviewer. I thank you a lot for the time spent in reading my paper and also thank you so much for your appreciation.
Reviewer 2 Report
Article title is X-SLIP to predict the triggering of rainfall-induced shallow landslides in real time: implementation of a new feature based on rainfall forecasts. Authors found that LEWS are typically based on data-driven methods, whose validity is not as general as the physically-based models (PBMs) and the effectiveness is often debatable. The purpose of this research is to investigate the prediction quality of the simplified PBM SLIP (implemented in X-SLIP platform) when stability analyses are performed with rain forecasts. X-SLIP was updated to handle the GRIB files (format for weather forecast). Four real-time predic-tions were simulated on some towns of the Emilia Apennines (northern Italy), involved in wide-16 spread soil slips on 5th April 2013; specifically, maps of factor of safety related to this event were derived assuming that X-SLIP had run 72h, 48h, 24h and 12h in advance.
I can recommend it for publication based on the quality of article but after addressing the following changes.
I recommend to edit the main hook of the study, as the first sentence of the introduction should have to modify with given studies (1,2) as “Global warming and climate change are directly affecting the pattern of precipitation in our planet (1,2)”
(1)Assessing the Impact of Long-Term ENSO, SST, and IOD Dynamics on Extreme Hydrological Events (EHEs) in the Kelani River Basin (KRB), Sri Lanka. Atmosphere 2023, 14, 79
(2)Extreme weather events risk to crop-production and the adaptation of innovative management strategies to mitigate the risk: A retrospective survey of rural Punjab, Pakistan
Please write the main research questions and structure of article at the end of introduction section.
Results have already written very well.
Please separate the section of conclusion from the discussion. In the section of discussion, please justify or compare the main findings of the study with previous literature.
Please write the conclusion section using the given format: Objectives; Methods; Main results; and Policy implications.
Author Response
Dear Reviewer, thank you very much for your time in reading and providing your constructive comments. Please find my point-by-point answers in the following.
- I recommend to edit the main hook of the study, as the first sentence of the introduction should have to modify with given studies (1,2) as “Global warming and climate change are directly affecting the pattern of precipitation in our planet (1,2)”
(1)Assessing the Impact of Long-Term ENSO, SST, and IOD Dynamics on Extreme Hydrological Events (EHEs) in the Kelani River Basin (KRB), Sri Lanka. Atmosphere 2023, 14, 79
(2)Extreme weather events risk to crop-production and the adaptation of innovative management strategies to mitigate the risk: A retrospective survey of rural Punjab, Pakistan
Response- Thank you so much for your minute observation and valuable comments. I have mentioned them in revised manuscript. See new Lines 26-27.
- Please write the main research questions and structure of article at the end of introduction section.
Response- As suggested, the main research questions, together with a better described structure of article, is included at the end of the Introduction. See new Lines 71-72 and 74-85.
- Results have already written very well.
Response- Thank you.
- Please separate the section of conclusion from the discussion. In the section of discussion, please justify or compare the main findings of the study with previous literature.
Response- As suggested by the Reviewer, a specific conclusion section has been added and in the new “Discussion” section, the main findings of the study are compared with previous literature. See new Sections 5 and 6.
- Please write the conclusion section using the given format: Objectives; Methods; Main results; and Policy implications.
Response- A new Conclusion section has been added following the format suggested by the Reviewer. See Section 6.
Reviewer 3 Report
I have reviewed the manuscript “X-SLIP to predict the triggering of rainfall-induced shallow landslides in real time: implementation of a new feature based on rainfall forecasts”. I think the manuscript is complete and well displayed. The authors present an extensive geological charectisation with high quality graphical information. I really appreciate the topic as it is of primary importance in the management of the danger associated with rainfall-induced shallow landslides. The results are comprehensive, interesting and easy to read.
Figure 5b. I advise the authors to insert the contourlines in the map to make it more complete and readable.
Moreover, the authors speak about landslides in general or soil slips, but these landslides types can have different kinematics. I would advise the author to integrate paragraph 3.1 with a more in-depth description of the landslide typology to which the model applies. I also recommend inserting a new image with photographs of the most representative landslides. In my opinion the work would acquire robustness to better understand the analyzed processes.
GENERAL COMMENTS
In general (although I recommend integrating with what is written above) I believe the paper is acceptable for publication in present form.
I can only congratulate the authors for the manuscript.
I wish you good work!
Author Response
Dear Reviewer, thank you very much for your time in reading the paper and providing your constructive comments. Please find my point-by-point answers in the following.
- Figure 5b. I advise the authors to insert the contourlines in the map to make it more complete and readable.
Figure 5b is now updated with contourlines, as suggested.
2. Moreover, the authors speak about landslides in general or soil slips, but these landslides types can have different kinematics. I would advise the author to integrate paragraph 3.1 with a more in-depth description of the landslide typology to which the model applies. I also recommend inserting a new image with photographs of the most representative landslides. In my opinion the work would acquire robustness to better understand the analyzed processes.
The SLIP model is able to capture with a good approximation the initiation mechanism of all the main types of rainfall-induced shallow landslides (debris slide, debris/earth flow). I have followed your suggestion adding new lines in Section 3.1 to clarify what are the main types of rainfall-induced landslides and the SLIP suitability to capture the initiation mechanism of all of them. Please see new Lines 211-215.
I have also added some photos in Figure 4, specifying that they are representative of the types of phenomena occurring in this area (See Line 221).
Reviewer 4 Report
This manuscript presents an interesting and potentially very useful research about implementing rainfall forecasts on the X-SLIP platform to predict the triggering of rainfall-induced shallow landslides in real time. The proposed approach makes very good sense and should be promising. As an earlier version of the platform/software has already been adopted by the Italian government, it is likely that the paper will have a direct impact on practice and society at large.
The article title is good, but it is a bit unclear to the readers what X-SLIP is and the current title seems to be a little too long. The author can consider either removing the name (as this will be mentioned in the abstract anyway) by slightly rearranging the wordings in the title, e.g.:
“Incorporating rainfall forecasts for real-time prediction of rainfall-induced shallow landslides”
Or the author may clarify what X-SLIP is, e.g. software, in the title. The title can be revised as:
“Incorporating rainfall forecast data in X-SLIP software to predict the triggering of rainfall-induced shallow landslides in real time” or alike.
The topic is part of a larger endeavour to develop a reliable landslide early warning system. This is a cutting edge topic in the field and is becoming more important due to global warming and climate change, making it harder to predict rainfall and landslide. The author has identified a key research problem and proposed a feasible solution to enhance prediction. The enhanced platform/software was then successfully applied for real-time predictions for a few towns in Italy.
The statement below that summarises the results in the abstract is a bit unclear:
“The results indicated that the predictions with forecasts are as accurate as the ones derived with rainfall recordings only. A similar agreement was seen also on a non-landslide day, when no landslide occurred despite rainfall peaks comparable to the triggering ones.”
Readers will be able to know the details by reading the main body of the paper, but it is always important to convey a clear message in the abstract. Does the author mean that the use of actual rainfall recordings is the benchmark, which is the best but impossible “prediction”, whilst the predictions based on the proposed method are as accurate as the benchmark? Also, the proposed method does not give false alarms when there was actually no landslide. The authors may consider enhancing the clarify of the statements but hopefully without lengthening the abstract much.
The proposed method/platform employs a physics-based model (PBM). Meanwhile, it is enhanced by rainfall forecast data. Is it correct to say that it is somehow a hybrid approach that employs both data-driven method and physics-based model? Or we may call it a data-driven or data-enhanced PBM? The author does not need to revise the paper according to this comment, but feel free to consider if it may enhance the clarity at certain locations of the manuscript.
Section 5 is stated as Discussion and Conclusion, whilst Section 6 is Conclusions. Please check if they are correct.
The English presentation and the formatting of the manuscript are very good. This reflects that the author is very serious about the work. This gives higher credibility to the work.
Author Response
Dear Reviewer, thank you very much for your time in reading the manuscript and providing your constructive comments. Please find my point-by-point answers in the following.
- The article title is good, but it is a bit unclear to the readers what X-SLIP is and the current title seems to be a little too long. The author can consider either removing the name (as this will be mentioned in the abstract anyway) by slightly rearranging the wordings in the title, e.g.: “Incorporating rainfall forecasts for real-time prediction of rainfall-induced shallow landslides” Or the author may clarify what X-SLIP is, e.g. software, in the title. The title can be revised as: “Incorporating rainfall forecast data in X-SLIP software to predict the triggering of rainfall-induced shallow landslides in real time” or alike.
Nice observation. I have decided to keep the second suggested title, replacing "software" with "platform", for coherence with previous works that I published on this topic. Final title: "Incorporating rainfall forecast data in X-SLIP platform to predict the triggering of rainfall-induced shallow landslides in real time"
2. The statement below that summarises the results in the abstract is a bit unclear: “The results indicated that the predictions with forecasts are as accurate as the ones derived with rainfall recordings only. A similar agreement was seen also on a non-landslide day, when no landslide occurred despite rainfall peaks comparable to the triggering ones.” Readers will be able to know the details by reading the main body of the paper, but it is always important to convey a clear message in the abstract. Does the author mean that the use of actual rainfall recordings is the benchmark, which is the best but impossible “prediction”, whilst the predictions based on the proposed method are as accurate as the benchmark? Also, the proposed method does not give false alarms when there was actually no landslide. The authors may consider enhancing the clarify of the statements but hopefully without lengthening the abstract much.
Also this comment is very constructive. Please see new lines 18-21 modified according to your suggestion.
3. The proposed method/platform employs a physics-based model (PBM). Meanwhile, it is enhanced by rainfall forecast data. Is it correct to say that it is somehow a hybrid approach that employs both data-driven method and physics-based model? Or we may call it a data-driven or data-enhanced PBM? The author does not need to revise the paper according to this comment, but feel free to consider if it may enhance the clarity at certain locations of the manuscript.
Thank you for this observation. Rainfall forecast were maybe derived through data-driven methods, but the Italian Air Force provided me the final elaborations of their COSMO model, as described in Section 2.2. I believe that in my analyses SLIP is basically a PBM, though running with forecasts (coming from other methods, but not developed by myself). I prefer not to include further comments on this topic to avoid misleading the readers. Hope that the reviewer will understand.
Round 2
Reviewer 2 Report
Satisfied with revision and have no further concerns.
Author Response
Dear Reviewer, thank you once more for your constructive comments.